# Validating TDP1 as an Inhibition Target for Lipophilic Nucleoside Derivative in Human Cells

**DOI:** 10.3390/ijms262010193

**Published:** 2025-10-20

**Authors:** Irina A. Chernyshova, Tatyana E. Kornienko, Nadezhda S. Dyrkheeva, Alexandra L. Zakharenko, Arina A. Chepanova, Konstantin E. Orishchenko, Nikolay N. Kurochkin, Mikhail S. Drenichev, Olga I. Lavrik

**Affiliations:** 1Institute of Chemical Biology and Fundamental Medicine, Siberian Branch of the Russian Academy of Sciences, 630090 Novosibirsk, Russia; chernyshova0305@gmail.com (I.A.C.); dyrkheeva.n.s@gmail.com (N.S.D.); a.zakharenko73@gmail.com (A.L.Z.);; 2Federal Research Centre Institute of Cytology and Genetics, Siberian Branch of the Russian Academy of Sciences, 630090 Novosibirsk, Russia; 3Engelhardt Institute of Molecular Biology, Russian Academy of Sciences, 119991 Moscow, Russiamdrenichev@mail.ru (M.S.D.)

**Keywords:** DNA repair, Tyrosyl-DNA phosphodiesterase 1 inhibitor, topotecan, TDP1 knockout, lipophilic nucleoside

## Abstract

Tyrosyl-DNA phosphodiesterase 1 (TDP1) is an important DNA repair enzyme and its functioning is considered as one of the possible reasons for tumor resistance to topoisomerase 1 (TOP1) poisons such as topotecan. Thus, TDP1 inhibitors in combination with topotecan may improve the effectiveness of anticancer therapy. TDP1 acts somehow in a phospholipase manner, depleting the phosphodiester bond between lipophilic tyrosine residue and 3′ end of DNA; therefore, lipophilic molecules bearing aromatic substituents can interact with TDP1 and even possess high inhibitory activity, which is evidenced by data from the literature. Previously, we identified lipophilic nucleoside derivative (compound **6d**, IC_50_ = 0.82 µM) as an effective inhibitor of the purified enzyme TDP1 that enhances the cytotoxic, DNA-damaging, and antitumor effects of topotecan. However, the role of TDP1 inhibition in this synergistic effect remained not fully understood. In the present study, we have tested the hypothesis of a TDP1-dependent mechanism of action for compound **6d**, showing that it sensitizes wild-type A549 lung cancer cells, but not TDP1 knockout cells, to the cytotoxic effects of topotecan. The sensitizing effect was absent in non-cancerous HEK293A cells regardless of TDP1 status. Additionally, we analyzed the effect of compound **6d** and topotecan on the expression level of *TOP1* and *TDP1* to determine whether the observed synergy was due to direct TDP1 inhibition and/or changes in regulation of these enzymes. The data obtained shows that compound **6d** did not affect *TDP1* gene expression level in HEK293A and A549 WT cells. Thus, compound **6d** most probably does not suppress the transcription or mRNA stability of TDP1, and the synergistic action of **6d** with topotecan is related to TDP1 inhibtion.

## 1. Introduction

Tyrosyl-DNA phosphodiesterase 1 (TDP1) is a eukaryotic DNA-repairing enzyme with broad substrate specificity, allowing the removal of various blocking lesions from 3′-DNA ends [1]. TDP1 is a promising molecular target in cancer therapy because it participates in the repair of many DNA adducts formed by chemotherapeutic drugs such as topotecan, etoposide, bleomycin, temozolomide [2,3]. Topotecan (Tpc) is a topoisomerase 1 (TOP1) inhibitor from the camptothecin group and can be used in clinical practice for the treatment of cervical, ovarian cancer [4], colon cancer, and small cell lung cancer [5]. Its antitumor effect is based on the ability to stabilize the covalent TOP1-DNA adduct formed during the catalytic cycle of the enzyme. The accumulation of covalently stabilized TOP1-DNA complexes (TOP1cc) can lead to cell death due to the formation of single- and double-strand breaks in DNA [6]. Double-strand breaks can occur, for example, when TOP1cc are within 10 base pairs on opposite strands of the DNA duplex [7], or when they arise near an already existing single-strand break on the opposite strand [8], as well as during collisions with replication forks [9]. The TOP1cc is the best-studied cellular substrate of TDP1 [10,11]. The removal of TOP1cc by TDP1 activity may decrease the antitumor effect of Tpc. Thus, the loss of TDP1 function leads to significant accumulation of DNA damage and enhances tumor cell sensitivity to camptothecin derivatives [12,13]. A homozygous mutation in the *TDP1* gene (His493Arg) underlies the hereditary neurological disorder spinocerebellar ataxia with axonal neuropathy (SCAN1) [14]. Cells with this mutation display an eightfold increase in DNA breaks upon camptothecin treatment compared to normal cells [15].

Protein levels and enzymatic activities of TOP1 and TDP1 have been reported to be higher in the tumor tissue of non-small cell lung cancer patients compared to normal tissue [16]. Meisenberg S. et al. proposed to use the TDP1/TOP1 protein ratio as a promising indicator of the response to topotecan in small cell lung cancer [17]. An increase in the level of TOP1 expression was associated with an enhanced sensitivity of olaparib-resistant cells to another camptothecin derivative irinotecan [18]. Perego P. et al. suggested the involvement of TDP1 in resistance to camptothecins as they observed higher levels of TOP1 and TDP1 in resistant cell systems compared to parental cells. They also found that elevated TDP1 expression protects colorectal cancer cells from camptothecin-induced cell death [19]. These data suggest that TDP1 is a promising molecular target in the development of combined therapies that may be useful for increasing the sensitivity of tumors to chemotherapy drugs and reducing the toxicity of therapy.

To date, TDP1 inhibitors have been discovered in a variety of chemical groups, including orthovanadates, tungstates, progesterone derivatives, piperidinyl sulfamides, benzopentathiepines, sulfonic acids, diamidines, bile acid derivatives, thieno[2,3-b]pyridines, indenoisoquinolines, coumarins, usnic acids, and adamantanes [20,21,22,23,24,25,26,27,28,29,30,31]. The Y. Pommier group is actively searching for compounds that inhibit the activity of TOP1, which are then tested for their ability to inhibit the activity of TDP1. For example, among the TOP1 inhibitors of the indenoisoquinoline class developed by these researchers, there are compounds that also have inhibitory activity against TDP1 [25,29,32]. The authors have also developed derivatives of 7-azaindenoisoquinolines as triple TOP1/TDP1/TDP2 inhibitors [33]. Our team, in collaboration with colleagues from other organizations, has developed several classes of non-toxic TDP1 inhibitors active in the nanomolar and micromolar range of concentration; some of them can enhance the cytotoxic effect of Tpc in cell lines of various types [20,34]. The enamine and hydrazinothiazole derivatives of usnic acid and the coumarin derivative were found to enhance the effectiveness of Tpc chemotherapy in mouse tumor models [34,35,36].

Previously, our group demonstrated a correlation between the lipophilicity of nucleoside compounds and their ability to inhibit TDP1. It was found that pyrimidine nucleosides modified with bulky lipophilic substituents, such as benzoyl or triphenylmethyl, exhibit inhibitory activity at submicromolar concentrations, while unmodified compounds are inactive. Calculations of the coefficient of distribution (logP) in a 1-octanol/water system showed that the introduction of benzoyl groups significantly increased the lipophilicity of the molecules, which is likely to facilitate their penetration through cell and nuclear membranes [37]. We discovered lipophilic nucleoside derivative (compound **6d**, IC_50_ = 0.82 µM) to be an effective inhibitor of the purified enzyme TDP1 that enhanced the cytotoxic and DNA-damaging effects on HeLa cells to Tpc and effectively increased the antitumor effect of Tpc in a mouse model of ascites carcinoma Krebs-2 in vivo [38]. However, it was not completely clear that this synergistic effect depends on inhibition of TDP1 with **6d**. In this study, based on the hypothesis that TDP1 is a cellular target for **6d**, we investigated the sensitizing effect of this compound on the cytotoxic/antiproliferative effect of Tpc on cells with normal and deficient TDP1 level, both in tumor and non-tumor cells. We found that this sensitizing effect was observed for wild-type A549 lung cancer cells, but not in TDP1 knockout cells, which speaks in favor of TDP1 as a general target for this inhibitor. However, in non-tumor cells HEK293A, the sensitizing effect of the TDP1 inhibitor was not observed, regardless of the TDP1 status. We also assessed the expression levels of *TOP1* and *TDP1* under the influence of Tpc, compound **6d**, and their combination to determine whether the sensitizing effect of **6d** in combination with Tpc is a direct consequence of inhibiting TDP1 or if compound **6d** affects regulation of gene expression of *TOP1* and *TDP1*, which may influence the level of proteins in the cells.

## 2. Results and Discussion

### 2.1. Generation of TDP1 Knockout Cell Lines

The expression of DNA repair enzymes is often increased in cancer cells, enabling effective removal of DNA damage and resulting in the resistance of malignancies to therapy [39]. Therefore, DNA repair enzyme inhibitors are considered as promising drugs [40,41,42,43]. Previously, we identified a TDP1 inhibitor among lipophilic nucleoside derivatives (compound **6d**, IC_50_ = 0.82 µM), which was able to increase the sensitivity of HeLa cells to topotecan, enhance the level of DNA damage induced by topotecan in vitro, and potentiate the antitumor effect of topotecan in vivo in a mouse Krebs-2 ascites carcinoma model [38]. However, despite the effective inhibition of the recombinant TDP1 enzyme, the exact mechanism of action of compound **6d** remains unknown.

Based on the hypothesis that enhancement of topotecan action occurs specifically through suppression of TDP1 function, synergy should not be observed in the absence of this enzyme. To test this hypothesis, we created a panel of TDP1 knockout (KO) cell clones based on the A549 (lung adenocarcinoma) cell line using the CRISPR-Cas9 method according to the scheme in Figure 1A. The A549 cell model was selected since one of the clinical indications for Tpc is lung cancer [44,45]. A549 cells were transfected with a pSpCas9(BB)-2A-GFP (PX458) plasmid expressing a guide RNA, SpCas9 nuclease, and EGFP fluorescent protein. EGFP-expressing cells were sorted one per well using a cell sorter. Four cell clones, presumably containing deletions in the region of the third exon, which is the first protein-coding exon in the *TDP1* gene (see scheme in Figure 1A), were obtained (named B5, B6, B10, and E3). The obtained clones were characterized by PCR, Western blot, RT-qPCR, and assessment of TDP1 activity in the whole-cell extracts.

Genomic DNA from cell clones was analyzed by PCR for the presence of the target deletions. It was shown that clone E3 did not contain the deletion (Figure 1B, lane 1), while the other clones displayed a PCR product corresponding to a truncated *TDP1* gene fragment with a deletion (Figure 1B, lanes 2, 3, 4 for B5, B6, and B10, respectively). However, we cannot conclude homozygosity of these cells as we also observed, in the gel, a PCR product with the same mobility as the full-length *TDP1* gene.

Because of the ambiguity of the PCR data, Western blot analysis was used to assess TDP1 protein levels in the obtained clones. It was shown that B5, B6, and B10 clones lack the target protein (Figure 1C, lanes 1, 2, and 3, respectively), while E3 displayed a band corresponding to TDP1 (Figure 1C, lane 4). Since clone E3 did not carry a *TDP1* deletion and expressed the target protein according to Western blot, it was removed from further study.

The enzymatic activity of TDP1 in whole-cell extracts of the A549 WT and TDP1-KO (B5, B6, and B10) lines was studied, using a model oligonucleotide with 3′-Black Hole Quencher 1 (BHQ1) as a substrate, which can be removed by TDP1. The products of the enzymatic reaction were then separated in PAGE. In contrast to control WT extracts and purified recombinant TDP1, all studied clones showed no band on the gel corresponding to the reaction product (Figure 1D, lanes 4, 5, and 6 for B5, B6, and B10, respectively).

Relative levels of *TDP1* expression in the A549 B5, B6, and B10 lines were also determined by RT-qPCR. All clones exhibited *TDP1* gene mRNA, but expression levels were reduced by 1.7, 7.5, and 2.1 times in B5, B6, and B10, respectively, compared to wild-type cells (Figure 1E). Altogether, these data suggest that in the obtained cell lines, CRISPR-CAS9-edited *TDP1* gene alleles produce a non-functional transcript that cannot be fully translated by the cell.

The literature reports that TDP1-deficient cells display hypersensitivity to camptothecin and its analogs, such as Tpc [12,13,46]. We therefore used the MTT test to analyze the relative metabolic activity of A549 TDP1-KO clones after treatment with various Tpc concentrations (Appendix A). All TDP1-KO clones were more sensitive to the TOP1 inhibitor than wild-type cells: CC_50_ for B5 and B10 clones was about three times lower than for wild-type A549 (152 ± 14 nM and 163 ± 27 nM vs. 457 ± 29 nM). The most sensitive was B6 clone, with a 15-fold CC_50_ decrease (28 ± 3 nM), compared to A549 WT.

Previously, we observed a significant decrease in metastases in mice with Lewis carcinoma upon administration of TDP1 inhibitors (usnic acid derivatives) [34,35,36]. Such antimetastatic effects could be due to both TDP1 inhibition and additional targets of usnic acid derivatives. Knockout of TDP1 in HEK293A cells was shown to decrease cell motility (wound healing), as revealed earlier. Transcriptomic analysis of HEK293A TDP1-KO cells found changes in genes related to cell adhesion, communication, and MAPK pathway, relevant to cancer progression [47]. Next, we compared the migration of A549WT and TDP1-KO cells using wound healing assay.

The wound closure rates of A549 TDP1-KO clones were not statistically different from wild-type A549 cells (Figure 2). So, loss of TDP1 may potentially affect cell adhesion, but this is also dependent on other factors. A full understanding requires further research into the molecular mechanisms of TDP1 action, its role in invasion/metastasis in vitro and in vivo, and correlation of TDP1 levels with prognosis, tumor stage, and response to therapy, which is outside the scope of this study.

Therefore, the A549 TDP1-KO clones B5, B6, and B10 were chosen for further work since they possess the *TDP1* gene deletion, do not express TDP1 protein, and their cell extracts lack TDP1-specific activity. These cell lines do not differ in proliferation rate from wild-type, but show hypersensitivity to Tpc.

### 2.2. The Effect of Compound **6d** on Topotecan Action in Cell Lines with TDP1 Deficiency

We examined whether compound **6d** could potentiate the effect of topotecan in A549 WT and TDP1-KO cell lines. The MTT assay demonstrated that compound **6d**, at non-toxic concentrations 5 and 10 µM (CC_50_ ≥ 20 µM, relative metabolic activity curves are shown in Appendix A), enhanced the cytotoxic effect of Tpc on A549 wild-type cells by 3- and 4-fold, respectively (Figure 3A, CC_50_ (Tpc + DMSO) 460 ± 30 nM, (Tpc + 6d 5 µM) 150 ± 10 nM, (Tpc + 6d 10 µM) 115 ± 15 nM). To rule out additive effects of the drug combinations, combinatorial indexes (CIs) were calculated for each Tpc/**6d** pair using the CompuSyn software version 1.0. according to the program manual [48,49]. CI values <1, =1, and >1 indicate synergism, additivity, and antagonism, respectively. Interestingly, CI < 1 and thus a synergistic effect was only seen across a Tpc concentration range of 120–5000 nM (Appendix A). These results highlight the potential of compound **6d** as a Tpc sensitizer, though further studies are needed to select the optimal compound ratio.

In the experiments on the A549 TDP1-KO lines, the relative metabolic activity curves following treatment with Tpc and its combination with **6d** were superimposed for all cell clones, i.e., no synergistic effect was observed. Results for clone B6 are shown as a representative example (Figure 3A), while data for B5 and B10 are presented in the Appendix A.

In addition to MTT, we also used the impedance-based real-time assay on the xCELLigence system [50,51]. We observed that the cell index (a value proportional to cell number and level of adhesion) of wild-type A549 cells treated with combination **6d** and Tpc decreased by 1.5-fold (for 5 and 10 µM of **6d**) and 2-fold (for 20 µM of **6d**) compared to using only Tpc (Figure 3B). This effect was not observed in TDP1-deficient A549 cells, supporting the MTT results (Figure 3B for clone B6 and Appendix A for clones B5 and B10).

The mechanism of Tpc action involves the trapping of TOP1 at DNA breaks generated by the enzyme’s own catalytic cycle. The resulting TOP1cc in dividing cells lead to accumulation of single- and double-stranded DNA breaks. Application of TDP1 inhibitors with Tpc may increase the number of such damages, resulting in more efficient tumor cell death.

Therefore, we determined the level of DNA damage in A549 WT and TDP1-KO cells treated with Tpc and its combination with **6d** and with the alkaline comet assay. Microphotographs of the slides are shown in Appendix A; the slides with cell control, solvent control, and compound **6d** are presented in Appendix A. It was shown that the percentage of DNA in the tail of A549 wild-type cells after individual or combined treatment with Tpc and compound **6d** did not significantly differ (18 ± 3% and 20 ± 4%, respectively, Figure 3C), in the case of using 1 µM concentration of **6d**, which is close to its IC_50_ (0.82 µM). This concentration of compound **6d** in the culture medium is probably insufficient to inhibit the TDP1 enzyme in cells. With increasing **6d** concentration above IC_50_ in combination with Tpc, a statistically significant (*p* < 0.01) increase in DNA damage was observed (Figure 3C): the % DNA in the tail reached 33 ± 3% for 5 µM and 39 ± 3% for 10 µM 577 vs. 18 ± 3% with Tpc monotherapy. In the same experiments for A549 TDP1-KO clones, Tpc treatment increased tail DNA up to 40% for all lines, which indicates an increased sensitivity to Tpc (Figure 3C for clone B6 and Appendix A for clones B5 and B10). However, **6d** had no effect on Tpc-induced DNA damage in these cells (Figure 3C for clone B6 and Appendix A for clones B5 and B10).

Thus, by three independent methods, we have demonstrated that compound **6d** is capable of enhancing the effect of Tpc on the A549 WT cell line. The synergistic effect of **6d** depends on the presence of TDP1, as all performed experiments showed no statistically significant enhancement of Tpc’s action for any of the three A549 TDP1-deficient cell lines. These data confirm that the molecular mechanism of **6d**/Tpc synergy is mediated specifically by inhibition or modulation of TDP1 activity.

We also studied whether compound **6d** could potentiate the effect of topotecan in human embryonic kidney HEK293A WT and TDP1-KO cell lines previously obtained and characterized in our laboratory [52]. The MTT test showed that **6d** at non-toxic concentrations (CC_50_ parameters are shown in Appendix A) did not change the relative metabolic activity of HEK293A WT or TDP1-KO cells treated with Tpc (Appendix A). Similar results were obtained via impedance-based real-time assay on the xCELLigence System: use of Tpc together with **6d** led to additive decreases in cell index (Appendix A). The synergistic effect was also not observed at the level of the amount of DNA damage according to the results of the comet assay experiments. The percentage of DNA in the tail of HEK293A WT cells was about 27% in case of treatment with Tpc or its combination with **6d** and these values did not differ statistically (Appendix A, slides suitable for cell control, solvent control, and compound **6d** are presented in Appendix A).

Absence of **6d** effect on Tpc cytotoxicity in non-tumor HEK293A WT cells may point to a selective action for this compound against tumor cells. Metabolism and DNA repair in HEK293A cells likely differ from those in A549 tumor cells; differences may involve TDP1 expression levels, post-translational enzyme modifications, or interactions with other repair complex proteins. Differences in **6d** membrane permeability or intracellular accumulation between HEK293A and A549 cannot be ruled out either. These results underscore the potential therapeutic value of **6d**, since selective action on tumor cells could mean a more favorable safety profile for clinical use. Nonetheless, further studies across more tumors and conditionally non-tumor cell lines are required to confirm this hypothesis.

### 2.3. Study of the Effect of Compound **6d** and Its Combination with Tpc on the Expression Levels of TDP1 and TOP1 Genes

The role of TOP1 is critical in oncogenesis, as rapid proliferation and active DNA replication in tumor cells create increased topological stress on the genome. As a consequence, cancer cells develop a greater dependence on TDP1, since this enzyme is crucial for repairing TOP1-induced DNA breaks. Jakobsen et al. demonstrated that both the concentration and activity of TDP1 and TOP1 proteins are significantly higher in tumor tissue of patients with non-small cell lung cancer than in normal tissues [16]. This may be due to a regulated feedback mechanism connecting TOP1 and TDP1 expression, or perhaps their expression is controlled by the same transcription factor. Perego P. et al. also suggested the role of TDP1 in the development of drug resistance, having observed increased levels of TOP1 and TDP1 proteins in two independently selected resistant cell systems versus their parental lines [19].

Therefore, in this work, we used RT-qPCR to assess the expression levels of *TDP1* and *TOP1* in HEK293A WT, A549 WT, and TDP1-KO cells in response to treatment with compound **6d**, Tpc, or the combination to understand how cells maintain genome stability and to attempt to explain the selective action of compound **6d** in tumor cells.

It was shown that compound **6d** does not affect *TDP1* gene expression in HEK293A and A549 WT cells (Figure 4). This suggests that compound **6d** does not suppress the transcription or mRNA stability of TDP1, and that its synergistic action with Tpc is related specifically to the inhibition of TDP1 enzyme activity.

We also observed that treatment by Tpc leads to reducing TOP1 mRNA levels by 5.5-fold in HEK293A WT cells and 2-fold in A549 WT cells, with TDP1 expression decreasing by 1.5- and 4-fold, respectively (Figure 4). The varying degrees of suppression of TOP1 and TDP1 in these two lines reflect their biological differences. For instance, A549 is highly proliferative and requires sustained function of TOP1/TDP1 for DNA replication and repair, so even moderate inhibition (2-fold for TOP1 and 4-fold for TDP1) can have a major biological impact. HEK293A cells are less dependent on TOP1/TDP1, making them more sensitive to TOP1 inhibition (5.5-fold decrease) but less reactive to changes in TDP1 (only 1.5-fold decrease). To assess the baseline levels of TDP1 and TOP1 proteins in A549 and HEK293A cells, we performed Western blot analysis and found that the TDP1 content was similar in both cell types, while the TOP1 content in non-tumor HEK293A cells was significantly higher than in tumor A549 cells (Appendix A). Moreover, wild-type A549 cells were significantly less sensitive to topotecan than HEK293A cells (CC_50_ values 450 μM versus 180 μM, respectively). Apparently, cell sensitivity to TOP1 inhibitors is determined not by the absolute expression levels of TOP1 and TDP1, but by their ratio, as suggested in the study [17].

Interestingly, this suppression of TOP1/TDP1 expression was not seen in A549 WT cells treated with the combination of Tpc and **6d** (expression levels of TDP1 and TOP1 in this group did not differ from untreated controls), though it was present for HEK293A WT (Figure 4). This may explain the observed synergy of compound **6d** and Tpc in the A549 WT but not in the HEK293A cell line. Theoretically, increasing TOP1 expression levels means the importance of this repair pathway in A549 cells. It could result in a certain range of more effective action of TOP1 poisons due to an increase in the number of protein targets, but simultaneously increasing TDP1 levels would not block the action of Tpc due to TDP1 inhibition by compound **6d**.

In the case of TDP1-deficient HEK293A clones, there were no significant changes in TOP1/TDP1 expression upon treatment with Tpc, compound **6d**, or their combinations (Appendix A). Based on the above, in this cell type Tpc, namely formed due to being trapped by poison TOP1-DNA adducts, apparently regulates TOP1/TDP1 expression only in the presence of TDP1. We hypothesize that TDP1 deficiency allows for activation of alternative repair pathways, such as TOP1cc removal by endonucleases [53,54,55].

When studying A549 TDP1-KO lines, distinct changes in TOP1/TDP1 expression were observed. For clone B5, TOP1 decreased 4-fold and TDP1 decreased 8.5-fold with Tpc treatment. Treatment with **6d** alone did not affect TOP1, but TDP1 was reduced by 1.8-fold. In combined treatment, TOP1 was unchanged, while TDP1 was reduced 2-fold (Figure 5). The data for B5 with Tpc resembled the wild-type, possibly indicating incomplete suppression of TDP1 expression—TDP1 mRNA may be truncated or nonfunctional, as the full-length protein was absent by Western blot.

For clone B6, TDP1 expression was found to be reduced 18-fold in all treatment types vs. untreated, with no significant change in TOP1 levels (Figure 5). Such strong downregulation suggests extreme instability of TDP1 transcript in this clone, possibly due to degradation mechanisms (e.g., via miRNA).

Clone B10 showed a 1.5-fold decrease in TOP1 and a 2-fold decrease in TDP1 upon treatment with any of the compounds (Figure 5). Here, a co-expression mechanism may be preserved, but responses to treatment are less pronounced than in wild-type. For all three A549 TDP1-KO clones, no restoration of TOP1/TDP1 expression occurred with combined Tpc and **6d** treatment, as seen in A549 WT cells. This may be due to TDP1 deficiency and stronger alternative repair pathways for TOP1cc functioning.

Thus, for three A549 TDP1-KO cell clones, similar patterns of changes in TOP1/TDP1 expression were observed. Clone B5 stood out somewhat, showing sharp decreases in TOP1/TDP1 with Tpc, as in wild-type (to a lesser extent in B10). B5 is likely an intermediate between the WT and B10 clones. A lack of TOP1 change and strong TDP1 transcript instability, along with minimal TDP1 expression (Figure 1), make B6 the most representative TDP1 knockout line.

Some differences between clones may result from epigenetic factors, differing levels of TDP1-deficiency compensation, or variations in CRISPR cassette integration. Our data also demonstrate the complex regulatory interplay between TOP1 and TDP1 and raise new questions about TOP1 regulation and cellular stress response mechanisms.

## 3. Materials and Methods

### 3.1. Cell Lines and Growth Conditions

The cell lines A549 (human lung adenocarcinoma) and HEK293A (human embryonic kidney 293) were obtained from the Russian Collection of Cell Cultures (RCCC) of the Institute of Cytology of the Russian Academy of Sciences (St. Petersburg, Russia). All cell lines used in this work were grown in DMEM-F12 medium (Thermo Fisher Scientific, Waltham, MA, USA) supplemented with 100 U/mL penicillin–streptomycin (ThermoFisher Scientific, Waltham, MA, USA), 1x L-alanyl-L-glutamine (GlutaMAX, Gibco, Waltham, MA, USA), and 10% FBS (ThermoFisher Scientific, Waltham, MA, USA), at 37 °C with 5% CO_2_ in a humidified atmosphere.

### 3.2. TDP1 Knockout A549 Clones

TDP1 knockout A549 clones were obtained as previously described for the HEK293FT cell line [51]. Briefly, we used pSpCas9(BB)-2A-GFP (PX458) plasmid (Addgene plasmid #48138) to obtain clones making deletion in the first protein-coding exon of the human *TDP1* gene. A549 cells were transfected with the pX458-TDP1-gRNA1 and pX458-TDP1-gRNA2 plasmids (0.25 µg each) using the Lipofectamine 3000 Reagent (ThermoFisher Scientific, Waltham, MA, USA). Forty-eight hours after transfection, GFP-positive cells were sorted using a BD FACSAria III Cell Sorter (BD Biosciences, Franklin Lakes, NJ, USA). Single-cell clones were expanded for two weeks, and genomic DNA was analyzed for CRISPR/Cas9-mediated deletions in the *TDP1* gene by PCR with the primers: TDP1-scF 5′-TCAGGAAGGCGATTATGGGAG-3′ and TDP1-scR 5′-TTGATGTGGAGGGCTCCAG-3′. The clones were further verified by Western blot, TDP1 enzyme activity assay, and RT-qPCR.

### 3.3. Western Blotting

For Western blot analysis, proteins from whole-cell extracts of HEK293A, A549 wild-type cells, and cell clones B5, B6, and B10 were separated by Laemmli electrophoresis in a 10% SDS-polyacrylamide gel and transferred onto a nitrocellulose membrane (TransBlot Turbo, Bio-Rad, Hercules, CA, USA) by the semidry transfer technique. The membrane was probed with a rabbit antibody to TDP1 (Invitrogen, Waltham, MA, USA), TOP1 (FineTest, Wuhan, China), or a rabbit antibody to β-actin (Sigma, St. Louis, MO, USA). Blots were then probed with a horseradish peroxidase-conjugated goat anti-rabbit IgG antibody (1:50,000 dilution, produced by L.E. Matveev, Biotechnological Laboratory, ICBFM SB RAS, Novosibirsk, Russia), and immunoreactivity was detected by chemiluminescence (West Pico PLUS, Thermo Fisher Scientific, Waltham, MA, USA) using the Amersham Imager 680 (GE Healthcare, Chicago, IL, USA).

### 3.4. TDP1 Gel-Based Enzyme Activity Assay

The activity of TDP1 was assessed using a 50 nM oligonucleotide substrate (5′-(5,6 FAM-AAC GTC AGG GTC TTC C-BHQ1)-3′), incubated with 10 nM recombinant human TDP1 or cell extract (1 mg/mL) at 37 °C for 30 min in buffer (50 mM Tris-HCl, pH 8.0, 50 mM NaCl, 7 mM β-mercaptoethanol). Reactions were stopped by adding gel loading buffer (TBE, 10% formamide, 7 M urea, 0.1% xylene cyanol, 0.1% bromophenol blue, 20 mM EDTA) and samples were heated to 95 °C for 5 min before loading. Products were analyzed by electrophoresis in a 20% denaturing polyacrylamide gel with 7 M urea. Gels were scanned using a Typhoon FLA 9500 (GE Healthcare, Chicago, IL, USA) and quantified using the QuantityOne 4.6.7 software (Bio-Rad, Hercules, CA, USA).

### 3.5. MTT Assay

The cytotoxicity of compounds was tested using the HEK293A, A549 wild-type, and TDP1-KO cell lines with the standard MTT test [56]. Cells were plated at 5000 cells/well. Test compounds were added the day after seeding (1:100 dilution to total medium volume; final DMSO 1%) and cultures were growth for 3 days. Control cells were cultured with 1% DMSO. Measurements were performed in triplicate. The combination index (CI) of Tpc and compound **6d** on the A549 WT line was calculated using CompuSyn software version 1.0. (ComboSyn, Paramus, NJ, USA) according to the manual [49].

### 3.6. xCELLigence Real-Time Cell Analysis (RTCA)

Compound cytotoxicity was evaluated with HEK293A WT, A549 WT, and TDP1-deficient (clones B5, B6, B10) cells on an xCELLigence Real Time Cell Analyzer (Agilent, Santa Clara, CA, USA) and analyzed with RTCA software 2.2.0 (Agilent, Santa Clara, CA, USA). A total of 5000 cells per well were seeded, and compounds were added at different concentrations (indicated in Figure 3, Appendix A) the next day. Control cells were treated with 0.1% DMSO. Monitoring was performed for several days in two parallel experiments.

### 3.7. Alkaline Comet Assay

The alkaline comet assay was conducted as previously described [38]. Briefly, cells were seeded in 24-well plates at 0.05 million/mL. The next day, cells were treated with the tested compounds and incubated for 2 h. Suspension was mixed with 1% molten low melting agarose (CertifiedTM LMAgarose; Bio-Rad, Hercules, CA, USA) and transferred to slides pre-coated with 1% normal melting agarose (Agarose; Helicon, Moscow, Russia), then solidified at 4 °C. Slides were incubated in lysing solution (2.5 M NaCl, 100 mM EDTA, 10 mM Tris base, 1% Triton, 5% DMSO, pH 10.0) for 1 h and in electrophoresis buffer (300 mM NaOH, 1 mM EDTA, pH > 13) for 45 min at 4 °C. Electrophoresis was at 20 V, 450 mA, for 10 min on ice. Slides were washed with cold water and stained with SYBR Green I (Thermo Fisher Scientific, Waltham, MA, USA). Images were obtained with a CELENA S digital microscope (Logos Biosystems, Inc., Annandale, VA, USA), analyzed with Comet analysis software version 1.0 (Trevigen, Inc., Gaithersburg, MD, USA). At least 500 cells per sample were analyzed. DNA damage was quantified as median % Tail DNA = 100 × (tail fluorescence/total comet fluorescence).

### 3.8. Wound Healing Assay

The wound healing assay was performed for A549 WT and TDP1-KO (B5, B6, B10) cells by real-time cell imaging using the Cell-IQ MLF system at the Institute of Cytology and Genetics SB RAS. A total of 4 × 10^5^ cells per well were seeded in a 12-well plate. After 24 h of incubation, the cell monolayer was scratched with a sterile lancet. Imaging was performed for approximately 24 h on the Cell-IQ MLF automated culture and analysis system (Chip-Man Technologies, Tampere, Finland). Data were processed using Cell-IQ Analyzer 4 Pro-Write Version AN4.3.0 software and OriginPro 8.6.0. software.

### 3.9. RT-qPCR

A549 WT, HEK293A WT, and TDP1-KO cells were grown at 1–2 million per well in a six-well plate for 20–21 h in duplicate. The next day, cells were treated with Tpc (1 µM + 1% DMSO), compound **6d** (10 µM), or their combination for 5 h. RNA was isolated with TRIzol (Thermo Fisher Scientific, Waltham, MA, USA) per manufacturer’s instructions, dissolved in water, and quantified with a Nanodrop 1000 spectrophotometer (Thermo Fisher Scientific, Waltham, MA, USA). RNA was treated with DNase I to remove DNA: 10 μg RNA with 2 units RNase-free DNase I and buffered (40 mM Tris-HCl pH 8.0, 10 mM MgCl_2_, 10 mM CaCl_2_) at 37 °C for 20 min; DNase I was inactivated with 5 mM EDTA at 75 °C for 10 min; RNA samples were stored at –70 °C.

RT–qPCR was performed using BioMaster RT–PCR SYBR Blue (×2) (Biolabmix, Novosibirsk, Russia). The 20 µL reaction included 10 ng RNA, primers, and enzyme mix. Reactions were run on a LigthCycler96 (Roche, Basel, Switzerland) with reverse transcription at 45 °C for 1800 s; preincubation was conducted at 95 °C for 300 s, and 36 cycles at 95 °C for 10 s, 60 °C for 10 s, and 72 °C for 10 s; signal was detected at 84 °C for 5 s. Primers for reference and target genes were designed using NCBI and synthesized at LBMC ICBFM SB RAS. Experiments were in triplicate; *GAPDH* and *B2M* were calibrators. Amplification efficiency, relative expression (ΔΔCt), and standard errors were calculated with LigthCycler96 software 1.1.0.1320 (Roche, Basel, Switzerland). Results are means ± SD. Primer sequences and amplifications are summarized in Table 1.

### 3.10. Statistical Analysis

Statistical processing of Comet assay and RT-qPCR data was carried out using one-way ANOVA (STATISTICA software version 12.5, TIBCO Software Inc., Palo Alto, CA, USA). Post hoc assessment was performed using Tukey’s honestly significant difference (HSD) test. Data with *p* < 0.01 were considered statistically significant. Pairwise comparison of cell survival at different concentrations of Tpc in the presence and absence of compound **6d** was performed using the Mann–Whitney test.

## 4. Conclusions

To test the hypothesis of a TDP1-dependent mechanism of action for previously found lipophilic nucleoside derivative TDP1 inhibitor compound **6d**, TDP1-knockout clones of the A549 cell line were generated using the CRISPR-Cas9 method. Three obtained clones (B5, B6, and B10) contained a deletion in the *TDP1* gene, did not express TDP1 protein, and did not display its specific enzymatic activity. They also did not differ from the wild-type cells in wound closure rate, indicating no effect of TDP1 on cell adhesion in A549 tumor cells.

We established, with MTT, xCELLigence, and Comet assay methods, that compound **6d**, which efficiently inhibits recombinant TDP1 enzyme, can potentiate the cytotoxic and DNA-damaging effect of Tpc in A549 WT cells. For TDP1 knockout clones, the synergistic effect of compound **6d** and Tpc was not observed in the absence of TDP1, indicating the importance of this enzyme in the mechanism of the examined TDP1 inhibitor at the cellular level. At the same time, different TDP1-KO clones demonstrated somewhat different response patterns, which supports the need to obtain a panel of single-gene knockout clones using CRISPR-Cas9. Interestingly, compound **6d** did not show synergistic action with Tpc in the non-tumor HEK293A cell line, unlike the A549 tumor cell line which may indicate selective action of the compound. Further studies on a wider range of tumor and conditionally normal cell lines are needed to confirm this hypothesis. Potentially, such selectivity opens new prospects for developing safer therapies based on the combined action of Tpc (TOP1 poison) and TDP1 inhibitors.

The obtained data indicate that compound **6d** enhances the action of Tpc specifically through the inhibition of TDP1 and emphasize the significance of this enzyme as a target for increasing the efficiency of cancer therapy with topoisomerase 1 inhibitor drugs.

## Figures and Tables

**Figure 1 ijms-26-10193-f001:**
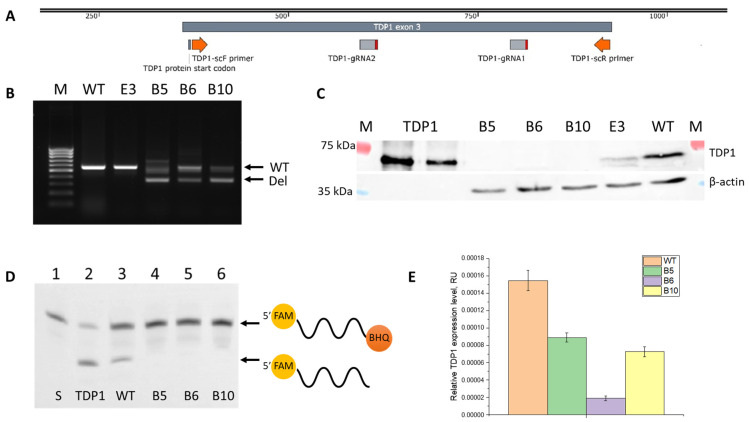
Creation and characterization of a panel of TDP1-KO clones based on the A549 cell line. (**A**) Scheme of CRISPR-Cas9-mediated deletion and primer locations for PCR. (**B**) Electropherogram of PCR results on genomic DNA from the clones. M—100 bp ladder (100–1000 bp). In B5, B6, and B10 clones (lanes 2, 3, and 4, respectively) a deletion product is present, while E3 matches wild-type (lane 1). (**C**) Western blot analysis of whole-cell extracts from A549 cells. In B5, B6, and B10 (lanes 1, 2, and 3, respectively), there is no TDP1 band. Lane 4: TDP1 present in E3 extract. (**D**) TDP1 activity in A549 WT and TDP1-KO extracts. No TDP1 activity in B5, B6, and B10 cell clones (lanes 4, 5, and 6). (**E**) Relative *TDP1* expression level in A549 WT and T DP1-KO cells measured by RT-qPCR. Expression was reduced by 1.7-, 7.5-, and 2.1-fold for clones B5, B6, and B10, respectively.

**Figure 2 ijms-26-10193-f002:**
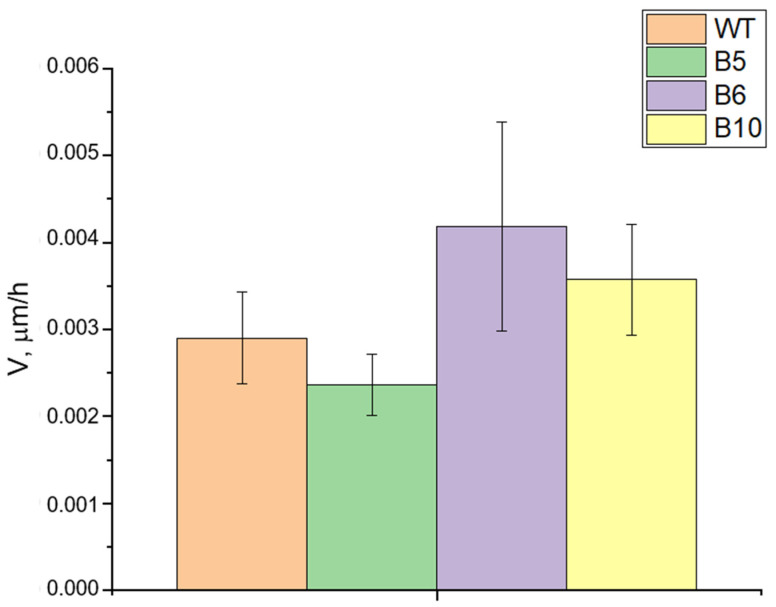
Histogram showing the wound closure rates for A549 WT and TDP1-KO cells (rate of closure, μm/h).

**Figure 3 ijms-26-10193-f003:**
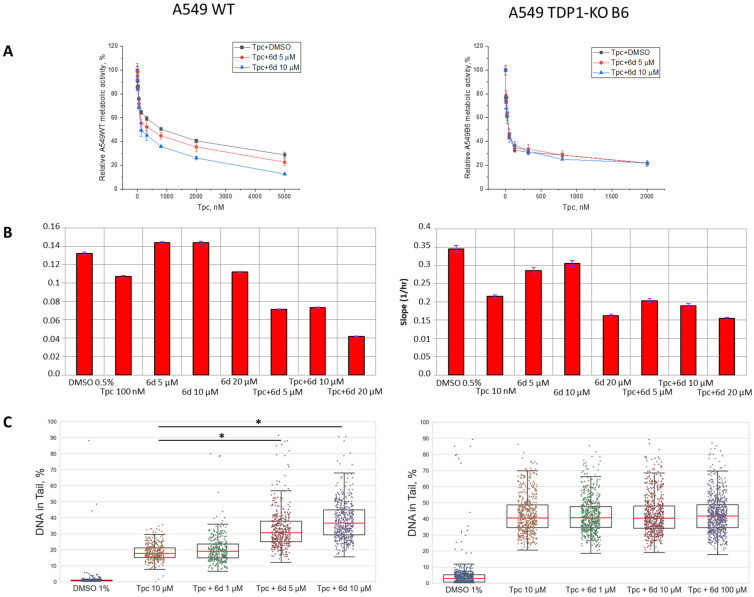
The effect of compound **6d** on the action of Tpc in A549 WT and TDP1-KO B6 cell lines. (**A**) MTT assay data. Combined Tpc and **6d** treatment decreased A549 WT relative cell metabolic activity (**left**) with no synergy in B6 cells (**right**). Differences in the presence and absence of the compound **6d** are significant according to the Mann–Whitney test at concentrations ranges of topotecan 320–5000 nM, *p* < 0.05. (**B**) Cell indexes obtained by impedance-based real-time xCELLigence assay. Combination of Tpc and **6d** reduced cell index compared to Tpc alone for A549 WT (**left**), but not B6 (**right**). (**C**) DNA damage assessed by Alkaline Comet assay. Percentage of DNA in tail is statistically significantly increased (* *p* < 0.01, Tukey’s HSD) after combined treatment in A549 WT (**left**) compared to Tpc alone. No effect was found in TDP1-KO B6 cells (**right**) (% tail DNA did not differ among groups).

**Figure 4 ijms-26-10193-f004:**
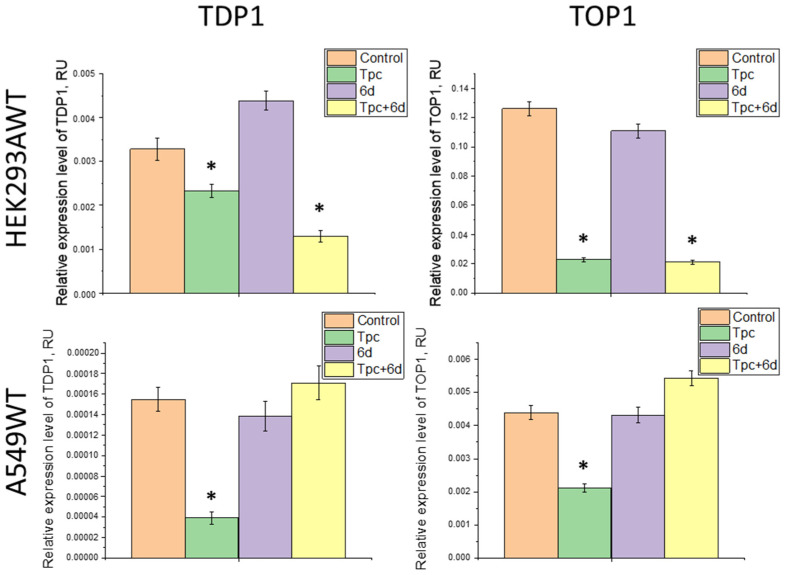
Relative expression levels of *TDP1* and *TOP1* in HEK293A and A549 WT cells under treatment by Tpc, compound **6d**, and their combination. Data labeled * is significantly different from the control *p* < 0.01, Tukey’s HSD.

**Figure 5 ijms-26-10193-f005:**
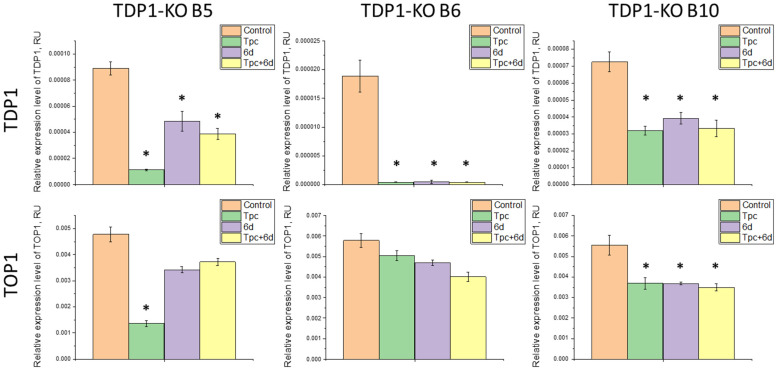
Relative expression levels of *TDP1* and *TOP1* in A549 TDP1-KO clones (B5, B6, B10) under the treatment with Tpc, compound **6d**, and their combination. Data labeled * is significantly different from the control *p* < 0.01, Tukey’s HSD.

**Table 1 ijms-26-10193-t001:** Primer sequences and amplification efficiency.

Gene	Primer Sequence	Amplification Efficiency for HEK293A Cells	Amplification Efficiency for A549 Cells
*GAPDH*	AGATCATCAGCAATGCCTCCT	2.01 ± 0.16	1.9 ± 0.2
	TGGTCATGAGTCCTTCCACG		
*B2M*	CGCTCCGTGGCCTTAGCTGT	1.95 ± 0.09	1.87 ± 0.15
	AAAGACAAGTCTGAATGCTC		
*TOP1*	CCTCCTGGACTTTTCCGTGG	2.0 ± 0.2	2.07 ± 0.13
	GGAACCTTGGCATCTTTGCTAC		
*TDP1*	AAGACATCTCTGCTCCCAATG	2.2 ± 0.3	2.2 ± 0.15
	TTCCCTTTATCCAGCATGTCC		

## Data Availability

The original contributions presented in this study are included in the article/Appendix A. Further inquiries can be directed to the corresponding author(s).

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
