# Peer review of "Validating TDP1 as an Inhibition Target for Lipophilic Nucleoside Derivative in Human Cells"

_ijms, 2025, doi:10.3390/ijms262010193_

Round 1

Reviewer 1 Report

Comments and Suggestions for Authors

The manuscript «Validating TDP1 as an Inhibition Target for lipophilic nucleoside derivative in human cells » by Chernyshova et al. is devoted to the role of TDP1 inhibition in the synergistic effect with topotecan on cancer cells. The authors confirmed the hypothesis of a TDP1-dependent mechanism of action in the cell for the compound 6d, previously found as an efficient inhibitor of the purified enzyme.

I recommend the paper for publication after minor revision.

  1. Introduction: The TOP1cc is the best-studied cellular substrate of TDP1 [10,11] that is why TDP1 activity may counteract the antitumor effect of Tpc – the authors should check this sentence and rewrite it.
  2. There is no information about combination index (CI) calculation.
  3. The authors should add an information about xCELLigence Real-Time Cell Analysis (RTCA) method, compounds concentrations etc.
  4. Results and Discussion: «this enzyme is crucial for repairing TOP1-induced DNA breaks and is the primary mechanism for restoring TOP1cc.» – there is no logic in this sentence.
  5. On Figures 4 and 5 WT should be changed to control.
  6. The authors should give СС50 for topotecan on HEK293A cells.

Author Response

The authors are grateful to the Reviewer for their suggestions and useful comments that helped improve the article. We carefully reviewed each comment and responded to all of them. Please find here the responses to your comments.

We appreciate your interest in our work. All the changes are marked in yellow in the main text.

  1. Introduction: The TOP1cc is the best-studied cellular substrate of TDP1 [10,11] that is why TDP1 activity may counteract the antitumor effect of Tpc – the authors should check this sentence and rewrite it.

Response 1: Indeed, the sentence doesn't seem entirely logical. We've corrected it as follows: “The TOP1cc is the best-studied cellular substrate of TDP1 [10,11]. The removal of TOP1cc by TDP1 activity may decrease the antitumor effect of Tpc.”

  1. There is no information about combination index (CI) calculation.

Response 2: We calculated the CI according to the program manual. We’ve added this sentence to the text.

  1. The authors should add an information about xCELLigence Real-Time Cell Analysis (RTCA) method, compounds concentrations etc.

Response 3: The concentrations of the compounds were selected individually for each compound and are shown in the figures. We added this information to the Materials and Methods section: “compounds were added at different concentrations (indicated in the Fig. 3, Fig. S3 and Fig. S7)”

  1. Results and Discussion: «this enzyme is crucial for repairing TOP1-induced DNA breaks and is the primary mechanism for restoring TOP1cc.» – there is no logic in this sentence.

Response 4: We have replaced the sentence with the following: “As a consequence, cancer cells develop a greater dependence on TDP1, since this enzyme is crucial for repairing TOP1-induced DNA breaks.”

  1. On Figures 4 and 5 WT should be changed to control.

Response 5: Corrected

  1. The authors should give СС50 for topotecan on HEK293A cells.

Response 6: We have added this information in subsection 2.3.

Reviewer 2 Report

Comments and Suggestions for Authors

In the manuscript titled “Validating TDP1 as an Inhibition Target for Lipophilic Nucleoside Derivative in Human Cells”, Chernyshova et al. investigate the role of tyrosyl-DNA phosphodiesterase 1 (TDP1) as a therapeutic target in enhancing the efficacy of topoisomerase I (TOP1) poisons. The authors build upon their previous identification of a lipophilic nucleoside derivative (compound 6d) as an effective TDP1 inhibitor, demonstrating that this compound sensitizes wild-type A549 lung cancer cells, but not TDP1 knockout cells, to the cytotoxic effects of topotecan. Importantly, this sensitizing effect was absent in non-cancerous HEK293A cells, suggesting a degree of tumor specificity. The study further shows that compound 6d does not alter TDP1 gene expression, indicating that its synergistic action with topotecan is due to direct enzymatic inhibition rather than transcriptional regulation.

Given the clinical importance of overcoming tumor resistance to TOP1 poisons, the findings presented here are timely and relevant, providing new insights into TDP1 inhibition as a strategy for potentiating anticancer therapy. However, before the manuscript can be considered for publication, several issues need to be addressed to improve clarity:

  1. The manuscript lacks a description of the statistical methods used. I could not find this information in the Methods section or figure legends, with the exception of Figure 3C, where it is stated that “percentage of DNA in tail is statistically significantly increased (p < 0.01).” However, the methods by which this significance was determined are not described. Without such details, it is not possible to assess whether the observed differences in MTT assays and qPCR assays (particularly in WT cells) are statistically significant. A clear description of the statistical approaches used is necessary.
  2. The finding that the sensitizing effect of compound 6d was absent in non-cancerous HEK293A cells regardless of TDP1 status is interesting, but it would be strengthened by providing direct comparisons of TDP1 and TOP1 expression levels between A549 WT and HEK293A WT cells. Such comparisons (mRNA and protein) could help clarify whether differences in expression levels contribute to the different responses. Western blot analysis in particular would add significant information to these conclusions, since the authors already have access to relevant antibodies. One possible explanation for the lack of sensitization in HEK293A cells could be higher baseline expression of TDP1, requiring higher compound concentrations for inhibition.
  3. The manuscript interprets both MTT and RTCA results as reflecting cell viability. However, neither assay directly distinguishes viability from proliferation arrest. To strengthen the conclusions, the authors should clarify their interpretation of these assays and, if possible, complement them with a simple viability assay such as trypan blue exclusion, which is inexpensive, and widely accessible.

Author Response

The authors are grateful to the Reviewers for their suggestions and useful comments that helped improve the article. We carefully reviewed each comment and responded to all of them. Please find here the responses to your comments.

The authors are grateful to the Reviewer for the kind words about our work and for constructive criticism. All the changes are marked in yellow in the main text.

  1. The manuscript lacks a description of the statistical methods used. I could not find this information in the Methods section or figure legends, with the exception of Figure 3C, where it is stated that “percentage of DNA in tail is statistically significantly increased (p < 0.01).” However, the methods by which this significance was determined are not described. Without such details, it is not possible to assess whether the observed differences in MTT assays and qPCR assays (particularly in WT cells) are statistically significant. A clear description of the statistical approaches used is necessary.

Response 1: Indeed, this was an oversight on our part. We added a subsection to the methods section describing the statistical processing methods used for the results, as well as to the figures and captions.

  1. The finding that the sensitizing effect of compound 6d was absent in non-cancerous HEK293A cells regardless of TDP1 status is interesting, but it would be strengthened by providing direct comparisons of TDP1 and TOP1 expression levels between A549 WT and HEK293A WT cells. Such comparisons (mRNA and protein) could help clarify whether differences in expression levels contribute to the different responses. Western blot analysis in particular would add significant information to these conclusions, since the authors already have access to relevant antibodies. One possible explanation for the lack of sensitization in HEK293A cells could be higher baseline expression of TDP1, requiring higher compound concentrations for inhibition.

Response 2: We performed a comparative Western blot analysis for TDP1 and TOP1 in A-549 and HEK293A cells and found that TDP1 levels were similar in both cell types, while TOP1 levels were significantly higher in HEK293A cells (Figure S11). It appears that sensitivity to topotecan depends not on the absolute expression levels of these proteins, but on the TDP1-TOP1 ratio (Meisenberg, C.; Ward, S.E.; Schmid, P.; El-Khamisy, S.F. TDP1/TOP1 Ratio as a Promising Indicator for the Response of Small Cell Lung Cancer to Topotecan. J Cancer Sci Ther 2014, 6, 258–267, doi:10.4172/1948-5956.1000280). The relationship between the presence/absence of sensitization and protein expression remains unclear. The relevant paragraph has been added to Results.

  1. The manuscript interprets both MTT and RTCA results as reflecting cell viability. However, neither assay directly distinguishes viability from proliferation arrest. To strengthen the conclusions, the authors should clarify their interpretation of these assays and, if possible, complement them with a simple viability assay such as trypan blue exclusion, which is inexpensive, and widely accessible.

Response 3: Thank you, we are sorry, you were wright. We replaced the term "viability" with "relative metabolic activity." 

Round 2

Reviewer 2 Report

Comments and Suggestions for Authors

I have reviewed the revised version of the manuscript “Validating TDP1 as an Inhibition Target for Lipophilic Nucleoside Derivative in Human Cells.” I appreciate the authors’ careful attention to the previous feedback. The additional details on statistical analysis and assay interpretation make the study much clearer and easier to follow.

The manuscript has improved noticeably in both clarity and presentation. I believe the authors have addressed all of my concerns thoughtfully, and the study now provides a well-supported contribution to the understanding of TDP1 inhibition and its potential therapeutic relevance.

I have no further comments and recommend the manuscript for publication.